# Inequalities in health system responsiveness among asylum seekers and refugees: A population-based, cross-sectional study in Germany

**Louise Biddle**[1,2]*, **Judith Wenner**[1], **Kayvan Bozorgmehr**[1,2]

**1** Department of Population Medicine and Health Services Research, School of Public Health, Bielefeld University, Bielefeld, Germany, **2** Department of General Practice and Health Services Research, Section for Health Equity Studies and Migration, University Hospital Heidelberg, Heidelberg, Germany

\* Louise.biddle@uni-bielefeld.de

## Abstract

Global migration has sparked renewed interest in Universal Health Coverage in high-income countries. However, quality of care has received little attention. This study uses the concept of responsiveness to study quality of care for asylum seekers and refugees (ASR) in Germany and identify inequalities among this group. We report results from a population-based, cross-sectional health monitoring survey in Germany's third-largest federal state using random sampling methods. Established instruments were used to measure responsiveness, health status and socio-demographic factors. Data were weighted and adjusted logistic regression models applied to identify inequalities related to health status, structural and socio-demographic factors. N = 344 survey participants were included in the analysis (response rate 39.2%). Combined responsiveness was 77% (95%CI: 68%; 83%) but varied between domains. Responsiveness was poor for individuals with symptoms of anxiety (OR 0.35, 95%CI 0.13,0.99), longstanding illness (OR:0.42, 95%CI:0.17,1.06) and diminished health-related quality of life (OR:0.24, 95%CI:0.06,0.95). Individuals from Southern Asia (OR: 0.24, 95%CI: 0.07,0.86) and young participants (OR:0.31, 95%CI:0.12,0.82) also reported less responsive care. Unique patterns of explanatory factors were identified within each responsiveness domain. We found important differences in responsiveness related to health, socio-demographic and structural factors, both in combined responsiveness and in individual domains. Inequalities related to health status factors are particularly concerning given the potential implications for equity of access. Future research should explore responsiveness for different sectors, include individuals who have not utilised healthcare and allow for the adjustment of differential expectations of care between population groups.

## 1. Introduction

The "summer of migration" in 2015 and its aftermath saw large numbers of asylum seekers and refugees (ASR) being displaced across the Middle East, Europe and elsewhere, with an estimated 25.9 million ASR globally in 2018 [1]. Germany alone received over 1.7 applications

**Data Availability Statement:** This research is based on sensitive human research participant data containing potentially identifying information. Participants have agreed that their data may be

used for non-profit research purposes by academic institutions. Data and survey instruments are available upon request through Respond. AMED@med.uni-heidelberg.de.

**Funding:** This study was funded by the German Federal Ministry for Education and Research (BMBF) in the scope of the project RESPOND (Grant Number: 01GY1611; Principal applicant: KB). The funders had no role in study design, data collection and analysis, decision to publish, or preparation of the manuscript.

**Competing interests:** The authors have declared that no competing interests exist.

for asylum from 2015 to 2018 [2], and although the numbers of new applicants have since declined it is still faced with the challenge of integrating these individuals into social and welfare systems. Health systems represent a crucial piece of infrastructure for newly arriving ASR, as increased exposure to risk factors before, during and after flight can leave them with a higher burden of illness [3]. Mental health, sexual and reproductive health, chronic illnesses, child and adolescent health and perinatal health in particular are issues which need to be addressed effectively and efficiently after arrival in a new country [4].

Holding countries accountable to the principles of Universal Health Coverage [5], the health systems literature has put much attention on legal entitlements to care and accessibility of health services for ASR. And rightly so: Across the European Union, 15/27 countries do not grant the same health entitlements to individuals seeking protection as are granted for the resident population [6]. In Germany, access to services is restricted to treating "acute illness and pain", as well as the provision of maternity care, for the first 18 months after arrival or when the asylum claim has been granted [7]. Even in those countries where full entitlements are formally granted, linguistic, bureaucratic, financial, cultural and structural barriers may impede access to essential services and result in inadequate provision of health services [8].

However, so far quality of care, as one of the three dimensions of Universal Health Coverage [5], has received little attention in health systems research on ASR [3]. The concept of responsiveness, defined as "aspects of the way individuals are treated and the environment in which they are treated during health system interactions" [9], may prove a useful tool in assessing quality of care for forced migrants. It was enshrined as one of the key outcomes of well-functioning health systems by the World Health Organization (WHO) in the World Health Report 2000 [10]. The inclusion of this concept was borne out of a growing recognition of the importance of the patients' experiences with health services for the success of the healthcare encounter: patients who have bad experiences in the doctor-patient relationship are less likely to adhere to the doctor's recommendations, take the prescribed medicines, or continue to engage with healthcare services [9]. WHO defines eight areas, or domains, in which responsiveness should be assessed: respectful treatment, prompt attention, communication, quality of basic amenities, confidentiality, choice, social support and autonomy. The dimension of communication was added by the WHO in later iterations of the responsiveness model, while the dimension of social support applies only to inpatient settings.

Responsiveness can be viewed as both an extrinsic and an intrinsic outcome of the health system [11]. It is an extrinsic outcome due to its strong links with the accessibility of care, increasing the first-time and continued use of health services by individuals in need of care, and thus potentially leading to improved health outcomes [12]. However, the responsiveness of the health system is an important outcome in its own right. Just as in other public sectors, the structures and systems put in place should reflect not only the goals as defined by institutional actors, but also the individuals which use and are dependent on these systems. The concept has close ties to existing frameworks of quality of care, representing the non-technical aspects of quality of care of the patient interaction, for example as laid out in Donabedian's quality of care model [13].

The ASR population is highly heterogeneous and has been characterised as being "super-diverse", owing to linguistic, socio-demographic and health differences as well as differential experiences of the flight process itself [14, 15]. Therefore, experiences of responsiveness may differ widely among this group. In order to identify potential shortcomings in the quality of care for particular patient groups, an understanding of differences in responsiveness among this heterogeneous group are essential. Mapping these inequalities can help us further understand whether and how the health system is currently serving patients from a quality perspective, and where further improvements are needed.

Keeping this in mind, the objective of the current analysis is twofold. First, we aim to assess the overall level of responsiveness of health services for ASR in Germany. Secondly, we aim to assess inequalities in responsiveness with regard to sociodemographic, structural and health status factors.

## 2. Materials and methods

### 2.1. Data collection

Data were collected in ASR reception centres and regional accommodation centres in Baden-Württemberg, Germany's third largest federal state, which received 17 055 first-time applications for asylum in 2021 [16]. Data were collected in 2018 using rigorously tested and translated questionnaires in nine languages. The questionnaire development process included pre-testing, cognitive interviews, and professional translation processes, which has been extensively described elsewhere [15]. The questionnaires covered items relating to socio-demographic characteristics, the asylum process, health status, healthcare utilization and quality of care, including the short-form WHO Responsiveness Instrument [9]. The questions were adjusted slightly in format in response to a qualitative, cognitive pre-test carried out prior to data collection [17]. Responsiveness questions were specifically aimed at healthcare received in Germany, but did not specify whether this was out-patient or in-patient care. The domain "Social Support" was omitted from the responsiveness questionnaire due to this joint assessment of both out-patient and in-patient care, resulting in a responsiveness instrument with a total of seven questions (see Box 1).

---

### Box 1. Questions included in the RESPOND questionnaire to capture responsiveness–answer options to all questions were "very good", "good", "moderate", "bad", "very bad" and "cannot say"

We are interested in hearing about your experience with healthcare services in Germany. We would like you to think about the last time you went to visit a doctor or another healthcare provider. How would you rate. . .

1. . . .the amount of time you waited at the doctor's before being attended to? -*timeliness domain*

2. . . .your experience of being greeted and talked to respectfully?—*respectful treatment domain*

3. . . .the experience of how clearly health care providers explained things to you? (Language and content easy to understand)–*communication domain*

4. . . .your experience of being involved in making decisions about your treatment?–*autonomy domain*

5. . . .the way health services ensured you could talk privately to health care providers?–*confidentiality domain*

6. . . .the freedom you had to choose your health care provider?–*choice domain*

7. . . .the cleanliness of the rooms inside the facility, including toilets–*cleanliness domain*

---

In accommodation centres, random sampling was carried out on the basis of 1938 accommodation centres, aiming to include 1% of the ASR population resident in state accommodation centres. Sampling was balanced on the number of ASR per region and the size of the accommodation centre. All residents in the chosen 65 accommodation centres were approached for data collection. For the data collection in reception centres, six out of nine facilities were sampled purposively to select large centres in a variety of administrative districts. Based on administrative lists of occupied rooms in each centre, 25% of rooms were randomly selected and all residents of these rooms approached for data collection. The sampling approach for reception and accommodation centres has been extensively described elsewhere [15, 18].

Data collection was carried out in a "door-to-door" approach, recruiting individuals personally with multi-lingual field teams [15, 18].

### 2.2. Data management

For this analysis, we included individuals with at least one response to one of the seven responsiveness questions. We excluded all individuals who indicated they had never utilised any health services (GP, specialist, dentist or psychologist), as the measurement of responsiveness specifically relates to a previous healthcare encounter. We treated all "cannot say" or "don't know" answers as missing.

The 5-point answer scale of responsiveness was dichotomised for each dimension, using "good" or "very good" answers as an indicator for responsiveness [9]. A combined responsiveness scale was calculated by averaging responsiveness scores across domains and setting the cut-off for a "responsive" rating for values averaging above 2.5 (analogous to a dichotomization at the cut-off between "good" and "moderate").

Selection of independent variables (Table 1) was guided by the available literature on factors influencing responsiveness, including socio-demographic variables such as age, sex, educational status, social status and country of origin [19–21] and health variables relating to physical and psychological well-being [22]. Education was included as an ordinal variable ranging from 1 (least educated) to 3 (most educated) based on educational attainment in school and on professional qualification (S1 Table). Measurement of subjective social status in Germany was based on an adapted version of the MacArthur Scale [23] and categorized to represent high, medium and low subjective social status [24]. To avoid issues with empty cells, data on nationality was grouped into regions based on the UN Geoscheme [25]. The three regions with the most participants (West Africa, South Asia and West Asia) where then included for analysis as binary variables. Age was included as a linear variable if it was treated as a covariate, and as a binary variable (30 years or younger/ 31 years or older) if it was treated as an independent variable in order to facilitate comparisons across factors. Health status variables covering general health status, health-related limitations, longstanding illnesses [26], Patient Health Questionnaire 2-item version (PHQ2) [27] and Generalised Anxiety Disorder 2-item version (GAD2) [28] were included as binary variables as reported in Table 1, except for health-related quality of life (HRQoL; EUROHIS-QOL) [29], which was divided into tertiles as no accepted thresholds for categorisation currently exist.

In addition to socio-demographic and health status differences in responsiveness, we also analysed structural factors relating to the specific situation of ASR, hypothesizing that conditions of the living environment and the asylum process may also influence the experience of responsiveness. These factors included residence status, type of accommodation (reception centre vs. regional accommodation centre) and urban-rural characteristics of the residential facility. Furthermore, ASR typically receive a health insurance card when a positive decision has been made on the asylum claim or after 18 months (15 months at the time of data collection),

**Table 1. Socio-demographic, health and structural factors included in the analysis of inequities of responsiveness.**

| | Variable | Categorisation | VIF |
|---|---|---|---|
| Socio-demographic factors | Gender | 0 = male | 1.10 |
| | | 1 = female | |
| | Age (binary) | 0 = over 30 years old | 1.17 |
| | | 1 = 30 years old and younger | |
| | Educational score | Score from 1 (least educated) to 3 (most educated) | 1.09 |
| | Subjective social status in Germany | 1 = low SSS | 1.12 |
| | | 2 = medium SSS | |
| | | 3 = high SSS | |
| | Region of origin: West Africa | 0 = not from West Africa | 1.60 |
| | | 1 = from West Africa | |
| | Region of origin: South Asia | 0 = not from South Asia | 1.69 |
| | | 1 = from South Asia | |
| | Region of origin: West Asia | 0 = not from West Asia | 1.55 |
| | | 1 = from West Asia | |
| | Number of close social contacts | 0 = no close social contacts | 1.22 |
| | | 1 = 1–2 close social contacts | |
| | | 2 = 3 or more close social contacts | |
| Health factors | Bad general health status | 0 = moderate/good/very good health | 1.49 |
| | | 1 = bad/very bad health | |
| | Health limitation | 0 = moderately/ not limited | 1.18 |
| | | 1 = severely limited | |
| | Chronic illness | 0 = no chronic illness | 1.44 |
| | | 1 = chronic illness | |
| | Quality of life | 1 = 1st tertile (worst quality of life) | 1.60 |
| | | 2 = 2nd tertile | |
| | | 3 = 3rd tertile (best quality of life) | |
| | Depression (PHQ2 score $\geq$ 3) | 0 = PHQ2 score $<$ 3 | 1.53 |
| | | 1 = PHQ2 score $\geq$ 3 | |
| | Anxiety (GAD2 score $\geq$ 3) | 0 = GAD2 score $<$ 3 | 1.65 |
| | | 1 = GAD2 score $\geq$ 3 | |
| Structural factors | Residence status | 1 = asylum seeker | 1.06 |
| | | 2 = asylum granted | |
| | | 3 = toleration/ rejection | |
| | Presence of a health insurance card | 0 = no health insurance card | 1.33 |
| | | 1 = health insurance card | |
| | Type of accommodation | 1 = regional accommodation centre | 1.35 |
| | | 2 = reception centre | |
| | Urban/ rural living environment | 0 = rural | 1.09 |
| | | 1 = urban | |

VIF = Variance Inflation Factor, PHQ2 = patient health questionnaire 2-item version, GAD2 = generalised anxiety disorder 2-item version

whichever comes first [7]. This has the potential to substantially reduce bureaucratic barriers in the health-seeking process, and therefore was also included as a structural factor.

## 2.3. Survey weighting

Adjustments for sampling frame were made by weighting for the probability of selection of each participant within the sampling design (S1 Text). The sample was calibrated using the

population characteristics of ASR from the years 2016, 2017 and 2018 [30], using sex, age group and region of origin variables. To enable calibration with a full data matrix, missing values were imputed using the single imputation technique in the R-package *mice* [31] (S1 Text).

## 2.4. Statistical analysis

For the purposes of logistic regression, missing values were imputed using multiple and multi-variate chained equations (Tables A and B in S1 Text), under the assumption that missing data are missing at random. Proportions of missing values per variable ranged from 0% (responsiveness: timeliness domain) to 29.4% (responsiveness: choice domain; Table B in S1 Text). The effect of imputation on the variance of each model was assessed using the fraction of missing information (FMI).

To address the first research aim, descriptive analysis was carried out without imputation of missing values. In the descriptive analysis, combined responsiveness scores were calculated only for those who had completed all questions of the responsiveness questionnaire (n = 207). Key sociodemographic, structural and health factors were tabulated for all included participants by sex. Proportions of unweighted and weighted responsiveness scores were calculated for each domain and for the overall responsiveness score with 95% confidence intervals. The design effect (DEFF) [32] was calculated to quantify the increase in variance due to weighting.

To address the second research aim, odds ratios (OR) and corresponding 95% confidence intervals (CI) for responsiveness were modelled using logistic regression on the fully weighted and imputed dataset. For the multiple logistic regression, combined responsiveness scores were calculated for all individuals included in the analysis (n = 344) using the fully imputed values. Combined responsiveness and each domain functioned as dependent variables in separate models. Independent variables representing key socio-demographic, health and structural factors (Table 1) were included in these models one at a time, adjusting for variables *apriori* defined as potential confounders, namely age (linear), sex and education. All independent variables were included in a final, multiple logistic regression model for combined responsiveness and each domain. Certainty of observed effects was assessed at the p≤0.05 significance level.

Multicollinearity was assessed for all independent variables on the fully imputed dataset, calculating the Variance Inflation Factor (VIF) and Condition Index (S1 Text). VIFs were low, ranging between 1.09 and 1.69 for independent variables (Table 1), while the Condition number was moderate at 28.14. No variables were excluded from the model due to multicollinearity.

Model fit was assessed using the model F-statistic, testing the hypothesis that all coefficients are equal to zero in each logistic model.

Both descriptive analysis and logistic regression were carried out in STATA version 15.

## 2.5. Ethics statement

Ethical approval was obtained from the ethics committee of the Medical Faculty Heidelberg on 12.10.2017 (S-516/2017). Participants were informed verbally and in writing about the aims of the study and the handling of their data. Participants gave consent to participation in the study by virtue of returning their questionnaires to us.

# 3. Results

## 3.1. Participants

A total of 560 ASR took part in the survey, representing a response rate of 39.2% (S1 Fig). Of these, n = 344 individuals provided accurate information on responsiveness and were included in the analysis.

Participants were predominantly male (66.1%) and young, with 47.3% of participants being under the age of 31. Educational status was varied, with just under half of participants (44.9%) reporting a medium educational status. The three most frequently reported regions of origin are Western Asia (26.4%), Southern Asia (26.8%) and Western Africa (20.4%). While most participants reported not having received a decision on their asylum application yet (59.1%), 61.7% of participants reported arriving in Germany over a year ago. The majority of participants (77.1%) reported living in collective accommodation centres, with the remaining participants living in reception facilities. While only 19.0% of participants reported being in bad or very bad health, a larger proportion reported longstanding illness (46.9%), symptoms of depression (PHQ2) (49.3%) and symptoms of anxiety (GAD2) (48.5%) (Table 2).

### 3.2. Responsiveness rating

The combined, weighted proportion of individuals reporting "(very) good" responsiveness was 77% (95%CI: 68%; 83%). However, ratings varied markedly between domains: while cleanliness and respect domains received high ratings of 86% (95%CI: 80%; 91%) and 85% (95%CI: 78%; 90%), respectively, responsiveness was much lower for choice (60%, 95%CI: 51%; 67%) and timeliness (52%, 95%CI: 46%; 59%) domains (Table 3).

Weighted and unweighted estimates do not appear to differ substantially, with differences on point estimates ranging from 0 percentage-points (timeliness domain) to 5 percentage-points (choice domain). The effects of weighting on the variance of estimates (DEFF) are moderate, ranging from 0.6-fold to 2.1-fold increases in variance when the sampling design is taken into account (Table 3).

### 3.3. Inequalities in combined responsiveness

Multiple logistic regression models show that poor health status is a predictor of low combined responsiveness once age, sex and educational status have been adjusted for. In particular, lower adjusted odds for "(very) good" responsiveness can be observed for individuals with a longstanding illness compared to those without a longstanding illness (OR: 0.47, 95%CI: 0.23; 0.98), for individuals with high PHQ2 scores compared to those with low PHQ2 scores (OR: 0.33, 95%CI: 0.16; 0.68) and for individuals with high GAD2 scores compared to those with low GAD2 scores (OR: 0.45, 95%CI: 0.11; 0.46). Furthermore, lower adjusted odds of "(very) good" responsiveness can be observed for those individuals with worse HRQoL compared to those with the best HRQoL (third quintile): an adjusted OR of 0.14 (95%CI:0.04; 0.51) for the first quintile, and an adjusted OR of 0.24 (95%CI: 0.07; 0.82) second quintile. A borderline significance can also be observed for the adjusted odds of "(very) good" responsiveness in individuals with a "(very) bad" health status compared to those in very good to moderate health (OR: 0.43, 95%CI: 0.17, 1.07) (S2 Table).

With regard to other factors, individuals from Southern Asia had lower adjusted odds of reporting "(very) good" combined responsiveness compared to individuals from other regions (OR:0.41, 95%CI: 0.19,0.89). However, no further substantial differences in combined responsiveness rating were found across other socio-demographic and structural variables in the models adjusted for *apriori* confounders (S3 and S4 Tables).

When all factors were included in the full model, lower adjusted odds for "(very) good" responsiveness could still be observed for those with longstanding illness (OR:0.42, 95% CI:0.17,1.06), those with a lower HRQoL (lowest tertile–OR:0.23, 95%CI:0.05,1.12; medium tertile–OR:0.24, 95%CI:0.06,0.95), those with high GAD2 scores (OR 0.35, 95%CI 0.13,0.99) and individuals from Southern Asia (OR: 0.24, 95%CI: 0.07,0.86). In the case of longstanding illness and the lowest HRQoL tertile, these effects were only borderline significant in the full

**Table 2. Sociodemographic characteristics of participants, by sex.**

| | | male | female | total |
|---|---|---|---|---|
| | | *n (%)* | *n (%)* | *n (%)* |
| **Age** | | | | |
| | 18–25 years | 65 (31.7%) | 28 (26.4%) | 93 (29.9%) |
| | 26–30 years | 40 (19.5%) | 14 (13.2%) | 54 (17.4%) |
| | 31–35 years | 34 (16.6%) | 24 (22.6%) | 58 (18.6%) |
| | 36–40 years | 29 (14.1%) | 18 (17%) | 47 (15.1%) |
| | 41+ years | 37 (18%) | 22 (20.8%) | 59 (19%) |
| | Total | 205 (100%) | 106 (100%) | 311 (100%) |
| **Education status** | | | | |
| | low | 42 (25.3%) | 33 (37.5%) | 75 (29.5%) |
| | medium | 77 (46.4%) | 37 (42%) | 114 (44.9%) |
| | high | 47 (28.3%) | 18 (20.5%) | 65 (25.6%) |
| | Total | 166 (100%) | 88 (100%) | 254 (100%) |
| **Region of origin** | | | | |
| | Eastern Europe | 5 (2.4%) | 6 (5.7%) | 11 (3.5%) |
| | Southern Europe | 5 (2.4%) | 10 (9.4%) | 15 (4.8%) |
| | Western Asia | 54 (26%) | 29 (27.4%) | 83 (26.4%) |
| | Southern Asia | 56 (26.9%) | 28 (26.4%) | 84 (26.8%) |
| | Western Africa | 48 (23.1%) | 16 (15.1%) | 64 (20.4%) |
| | Central Africa | 6 (2.9%) | 2 (1.9%) | 8 (2.5%) |
| | Northern Africa | 1 (0.5%) | 0 (0%) | 1 (0.3%) |
| | other | 33 (15.9%) | 15 (14.2%) | 48 (15.3%) |
| | Total | 208 (100%) | 106 (100%) | 314 (100%) |
| **Residence status** | | | | |
| | asylum seeker | 118 (60.2%) | 54 (56.8%) | 172 (59.1%) |
| | asylum granted | 41 (20.9%) | 19 (20%) | 60 (20.6%) |
| | toleration/rejection | 37 (18.9%) | 22 (23.2%) | 59 (20.3%) |
| | Total | 196 (100%) | 95 (100%) | 291 (100%) |
| **Months since arrival** | | | | |
| | 0–6 months | 46 (23.8%) | 24 (25.5%) | 70 (24.4%) |
| | 6–12 months | 25 (13%) | 15 (16%) | 40 (13.9%) |
| | 13–15 months | 43 (22.3%) | 21 (22.3%) | 64 (22.3%) |
| | 16–24 months | 68 (35.2%) | 26 (27.7%) | 94 (32.8%) |
| | 24–36 months | 11 (5.7%) | 8 (8.5%) | 19 (6.6%) |
| | Total | 193 (100%) | 94 (100%) | 287 (100%) |
| **Number of close personal contacts** | | | | |
| | none | 67 (37.9%) | 21 (23.6%) | 88 (33.1%) |
| | 1 or 2 | 72 (40.7%) | 53 (59.6%) | 125 (47%) |
| | 3 or more | 38 (21.5%) | 15 (16.9%) | 53 (19.9%) |
| | Total | 177 (100%) | 89 (100%) | 266 (100%) |
| **Urban/ rural setting** | | | | |
| | rural | 53 (24.5%) | 13 (11.7%) | 66 (20.2%) |
| | urban | 163 (75.5%) | 98 (88.3%) | 261 (79.8%) |
| | Total | 216 (100%) | 111 (100%) | 327 (100%) |
| **Accommodation type** | | | | |
| | regional accomodation centre | 168 (77.8%) | 84 (75.7%) | 252 (77.1%) |
| | reception centre | 48 (22.2%) | 27 (24.3%) | 75 (22.9%) |

(*Continued*)

**Table 2.** (Continued)

| | | male | female | total |
|---|---|---|---|---|
| | | *n (%)* | *n (%)* | *n (%)* |
| | Total | 216 (100%) | 111 (100%) | 327 (100%) |
| **health insurance card holder** | | | | |
| | no | 81 (42.4%) | 40 (39.6%) | 121 (41.4%) |
| | yes | 110 (57.6%) | 61 (60.4%) | 171 (58.6%) |
| | Total | 191 (100%) | 101 (100%) | 292 (100%) |
| **subjective social status in Germany** | | | | |
| | low | 124 (70.1%) | 67 (76.1%) | 191 (72.1%) |
| | medium | 31 (17.5%) | 15 (17%) | 46 (17.4%) |
| | high | 22 (12.4%) | 6 (6.8%) | 28 (10.6%) |
| | Total | 177 (100%) | 88 (100%) | 265 (100%) |
| **general health status** | | | | |
| | very good-medium health | 160 (81.2%) | 87 (80.6%) | 247 (81%) |
| | bad/very bad health | 37 (18.8%) | 21 (19.4%) | 58 (19%) |
| | Total | 197 (100%) | 108 (100%) | 305 (100%) |
| **longstanding illness** | | | | |
| | no | 106 (54.6%) | 55 (50.5%) | 161 (53.1%) |
| | yes | 88 (45.4%) | 54 (49.5%) | 142 (46.9%) |
| | Total | 194 (100%) | 109 (100%) | 303 (100%) |
| **Health limitation** | | | | |
| | not severely limited | 159 (84.1%) | 81 (78.6%) | 240 (82.2%) |
| | severely limited | 30 (15.9%) | 22 (21.4%) | 52 (17.8%) |
| | Total | 189 (100%) | 103 (100%) | 292 (100%) |
| **Health-related quality of life** | | | | |
| | 1st tertile (worst HRQoL) | 57 (31.7%) | 39 (39.4%) | 96 (34.4%) |
| | 2nd tertile | 70 (38.9%) | 28 (28.3%) | 98 (35.1%) |
| | 3rd tertile (best HRQoL) | 53 (29.4%) | 32 (32.3%) | 85 (30.5%) |
| | Total | 180 (100%) | 99 (100%) | 279 (100%) |
| **PHQ2 (depression)** | | | | |
| | negative | 100 (51.3%) | 49 (49.5%) | 149 (50.7%) |
| | positive | 95 (48.7%) | 50 (50.5%) | 145 (49.3%) |
| | Total | 195 (100%) | 99 (100%) | 294 (100%) |
| **GAD2 (anxiety)** | | | | |
| | negative | 106 (53.8%) | 46 (46.9%) | 152 (51.5%) |
| | positive | 91 (46.2%) | 52 (53.1%) | 143 (48.5%) |
| | Total | 197 (100%) | 98 (100%) | 295 (100%) |

HRQoL = health-related quality of life, PHQ2 = patient health questionnaire 2-item version, GAD2 = generalised anxiety disorder 2-item version

model. While no substantial effects are observed for individuals with "(very) bad" general health or those with a high PHQ2 score in the full model, lower odds for "(very) good" responsiveness could be observed in those participants under 31 years of age (OR:0.31, 95% CI:0.12,0.82) once all other factors had been adjusted for (Table 4).

### 3.4. Inequalities across domains

Results from fully adjusted logistic regression models for individual domains diverted partly from the results of combined responsiveness (Table 4).

**Table 3. Weighted and unweighted responsiveness results by domain, with 95% confidence intervals and overall design effect.**

| Domain | Responsiveness *unweighted* (95% CI) | Responsiveness *weighted* (95% CI) | Design effect (DEFF) |
|---|---|---|---|
| Timeliness | 0.52 | 0.52 | 1.59 |
| (n = 344) | (0.47; 0.57) | (0.46; 0.59) | |
| Respect | 0.82 | 0.85 | 2.20 |
| (n = 331) | (0.78; 0.86) | (0.78; 0.90) | |
| Communication | 0.61 | 0.63 | 2.36 |
| (n = 332) | (0.56; 0.66) | (0.54; 0.71) | |
| Autonomy | 0.63 | 0.66 | 1.82 |
| (n = 287) | (0.57; 0.68) | (0.58; 0.73) | |
| Confidentiality | 0.72 | 0.74 | 1.72 |
| (n = 291) | (0.67; 0.77) | (0.67; 0.80) | |
| Choice | 0.55 | 0.60 | 1.59 |
| (n = 243) | (0.49; 0.61) | (0.51; 0.67) | |
| Cleanliness | 0.83 | 0.86 | 1.76 |
| (n = 327) | (0.79; 0.87) | (0.80; 0.91) | |
| Combined | 0.73 | 0.77 | 3.06 |
| (n = 207) | (0.67; 0.79) | (0.68; 0.83) | |

CI = Confidence Interval

With regard to the health factors, significantly lower adjusted odds of reporting "(very) good" responsiveness were observed for participants with a high GAD2 score across the domains of timeliness (OR:0.33, 95%CI: 0.16,0.69) and autonomy (OR: 0.31, 95%CI: 0.12,0.86), with a borderline significant effect for the domain choice (OR: 0.38, 95%CI: 0.15,1.01). Individuals with a longstanding illness had lower adjusted odds of "(very) good" responsiveness in the confidentiality domain (OR:0.40, 95%CI:0.16,0.99), while individuals in the second HRQoL tertile had lower adjusted odds of "(very) good" responsiveness in the choice domain (OR: 0.26, 95%CI: 0.07,0.99). In the communication domain, participants with a health limitation report higher adjusted odds of "(very) good" responsiveness (OR: 3.84, 95% CI: 1.63,9.10).

Among the socio-demographic factors, significantly lower adjusted odds of "(very) good" responsiveness can be observed among participants under 31 years old across the domains of respect (OR: 0.17, 95%CI: 0.05,0.61) and confidentiality (OR: 0.46, 95%CI:0.22,0.98). For participants with a South Asian nationality, lower adjusted odds of "(very) good" responsiveness can be observed in the communication domain (OR:0.29, 95%CI: 0.11,0.81). Participants with a West African nationality (OR:4.44, 95%CI:1.29,15.37) and female participants (OR: 2.50, 95%CI:1.21,5.17) have higher adjusted odds of "(very) good" responsiveness in the autonomy domain. Finally, participants with smaller social network have lower adjusted odds of "(very) good" responsiveness in the respect domain (no close contacts–OR: 0.05, 95%CI: 0.01,0.42, 1–2 close contacts–OR:0.13, 95%CI: 0.02,0.95).

Although structural factors did not show effects on responsiveness in the combined model, residence status and type of accommodation demonstrate important effects in individual domains. Residents in reception centres have lower adjusted odds of "(very) good" responsiveness in the choice (OR:0.23, 95%CI:0.08,0.66) and cleanliness (OR:0.35, 95%CI:0.12,1.01) domains when compared to individuals in regional accommodation centres. Participants with an asylum rejection or a toleration have lower adjusted odds of "(very) good" responsiveness

**Table 4. Outcomes of multiple logistic regression models for combined responsiveness and each domain, including all potential socio-demographic, health and structural factors.**

| | | combined responsiveness | timeliness | respect | confidentiality | choice | communication | autonomy | cleanliness |
|---|---|---|---|---|---|---|---|---|---|
| | Average degrees of freedom | 25.958 | 29.899 | 25.944 | 29.584 | 27.132 | 28.520 | 27.451 | 29.127 |
| | Model F-value | 1.953 | 2.264 | 2.806 | 1.530 | 2.466 | 3.822 | 2.951 | 1.370 |
| | Model p-value (F-test) | 0.038 | 0.015 | 0.004 | 0.129 | 0.010 | <0.001 | 0.003 | 0.199 |
| | Maximum FMI | 0.416 | 0.250 | 0.460 | 0.301 | 0.325 | 0.411 | 0.374 | 0.373 |
| | | *Odds ratios (95% confidence interval)* | | | | | | | |
| Socio-demographic factors | Sex female (ref: sex male) | 0.61 (0.27,1.41) | 0.83 (0.42,1.65) | 0.60 (0.24,1.51) | 0.87 (0.40,1.88) | 0.56 (0.26,1.22) | 0.95 (0.46,1.95) | 2.50 (1.21,5.17)* | 0.91 (0.41,2.08) |
| | Age <31 years (ref: age 31+ years) | 0.31 (0.12,0.82) * | 0.83 (0.42,1.65) | 0.17 (0.05,0.61) ** | 0.46 (0.22,0.98) * | 0.40 (0.15,1.14) | 0.97 (0.40,2.36) | 0.80 (0.31,2.06) | 0.42 (0.13,1.32) |
| | Medium educational score (ref: lowest educational score) | 0.69 (0.23,2.13) | 0.42 (0.18,0.99) * | 0.50 (0.17,1.54) | 2.39 (0.90,6.36) | 1.31 (0.38,4.53) | 0.66 (0.27,1.64) | 1.50 (0.57,3.96) | 2.42 (0.75,7.88) |
| | Highest educational score (ref: lowest educational score) | 0.56 (0.17,1.91) | 0.61 (0.27,1.43) | 0.29 (0.08,1.08) | 2.31 (0.73,7.40) | 1.09 (0.32,3.79) | 0.95 (0.32,2.82) | 0.80 (0.21,3.07) | 1.50 (0.42,5.46) |
| | Medium SSS in Germany (ref: low SSS in Germany) | 0.54 (0.16,1.82) | 0.62 (0.26,1.52) | 0.92 (0.22,3.99) | 0.42 (0.11,1.61) | 0.59 (0.14,2.53) | 1.17 (0.39,3.54) | 1.24 (0.33,4.70) | 1.92 (0.37,9.92) |
| | High SSS in Germany (ref: low SSS in Germany) | 0.59 (0.15,2.35) | 0.66 (0.21,2.06) | 0.40 (0.09,1.94) | 1.20 (0.29,4.95) | 0.48 (0.08,2.83) | 0.41 (0.09,2.01) | 1.25 (0.33,4.79) | 0.41 (0.11,1.55) |
| | West Asian nationality (ref: other nationalities) | 0.68 (0.21,2.25) | 0.77 (0.31,1.91) | 2.06 (0.50,8.59) | 1.39 (0.64,3.04) | 1.23 (0.49,3.13) | 0.46 (0.13,1.59) | 1.51 (0.60,3.84) | 3.04 (0.79,11.81) |
| | South Asian nationality (ref: other nationalities) | 0.24 (0.07,0.86) * | 1.37 (0.47,4.03) | 0.60 (0.14,2.61) | 0.68 (0.26,1.81) | 0.36 (0.11,1.18) | 0.29 (0.11,0.81) * | 0.71 (0.28,1.82) | 1.27 (0.32,5.04) |
| | West African nationality (ref: other nationalities) | 2.05 (0.46,9.11) | 1.68 (0.60,4.76) | 3.41 (0.81,14.51) | 1.51 (0.39,5.83) | 3.04 (0.70,13.26) | 1.69 (0.52,5.51) | 4.44 (1.29,15.37) * | 1.68 (0.44,6.51) |
| | no close personal contacts (ref: 3+ close personal contacts) | 0.31 (0.07,1.50) | 0.65 (0.18,2.39) | 0.05 (0.01,0.42) ** | 1.19 (0.28,5.07) | 1.15 (0.34,3.91) | 0.40 (0.10,1.62) | 0.64 (0.17,2.42) | 0.67 (0.11,4.19) |
| | 1–2 close personal contacts (ref: 3+ close personal contacts) | 0.54 (0.13,2.26) | 0.47 (0.16,1.45) | 0.13 (0.02,0.95) * | 2.43 (0.74,8.01) | 1.30 (0.46,3.65) | 1.00 (0.33,3.03) | 0.65 (0.20,2.20) | 0.41 (0.09,1.86) |

*(Continued)*

**Table 4.** (Continued)

| | | combined responsiveness | timeliness | respect | confidentiality | choice | communication | autonomy | cleanliness |
|---|---|---|---|---|---|---|---|---|---|
| Health factors | Bad/ very bad general health | 1.03 (0.32,3.38) | 1.01 (0.45,2.29) | 1.40 (0.35,5.62) | 1.14 (0.33,3.99) | 0.48 (0.17,1.42) | 0.59 (0.19,1.87) | 0.57 (0.16,2.05) | 0.96 (0.35,2.66) |
| | (ref: moderate-v. good general health) | | | | | | | | |
| | Health-related limitation | 1.03 (0.33,3.23) | 2.28 (0.86,6.08) | 1.97 (0.58,6.72) | 0.90 (0.35,2.35) | 1.84 (0.63,5.40) | 3.84 (1.63,9.10) ** | 1.23 (0.49,3.11) | 0.80 (0.22,2.93) |
| | (ref: no health-related limitation) | | | | | | | | |
| | Longstanding illness | 0.42 (0.17,1.06) | 1.29 (0.52,3.23) | 0.36 (0.11,1.18) | 0.40 (0.16,0.99) * | 0.92 (0.39,2.21) | 0.98 (0.44,2.18) | 0.64 (0.28,1.50) | 0.65 (0.25,1.75) |
| | (ref: no longstanding illness) | | | | | | | | |
| | Lowest HRQoL tertile | 0.23 (0.05,1.12) | 0.49 (0.18,1.38) | 0.41 (0.07,2.31) | 0.43 (0.10,1.93) | 0.29 (0.05,1.84) | 0.44 (0.15,1.38) | 0.71 (0.17,2.95) | 0.81 (0.21,3.23) |
| | (ref: highest HRQoL tertile) | | | | | | | | |
| | Medium HRQoL tertile | 0.24 (0.06,0.95) * | 0.43 (0.17,1.13) | 0.28 (0.07,1.28) | 0.43 (0.10,1.93) | 0.26 (0.07,0.99) * | 0.47 (0.19,1.19) | 0.52 (0.19,1.48) | 0.43 (0.11,1.66) |
| | (ref: highest HRQoL tertile) | | | | | | | | |
| | Positive PHQ2 (ref: negative PHQ2) | 0.72 (0.27,1.93) | 0.87 (0.35,2.22) | 0.55 (0.18,1.66) | 1.03 (0.35,3.08) | 0.71 (0.22,2.30) | 0.73 (0.38,1.40) | 1.25 (0.51,3.10) | 0.36 (0.10,1.28) |
| | Positive GAD2 (ref: negative GAD2) | 0.35 (0.13,0.99) * | 0.33 (0.16,0.69) ** | 0.59 (0.19,1.90) | 0.43 (0.16,1.22) | 0.38 (0.15,1.01) | 0.58 (0.28,1.23) | 0.31 (0.12,0.86) * | 1.23 (0.37,4.06) |
| Structural factors | Residence status: asylum granted | 0.48 (0.15,1.62) | 0.61 (0.26,1.43) | 0.96 (0.27,3.48) | 0.58 (0.20,1.74) | 0.56 (0.18,1.75) | 0.66 (0.27,1.64) | 0.68 (0.21,2.21) | 0.43 (0.13,1.50) |
| | (ref: asylum seeker) | | | | | | | | |
| | Residence status: asylum rejected/ toleration (ref: asylum seeker) | 0.61 (0.19,2.00) | 0.54 (0.27,1.10) | 0.60 (0.15,2.46) | 0.81 (0.29,2.23) | 0.34 (0.12,1.02) | 0.41 (0.17,1.01) | 0.71 (0.25,2.00) | 0.61 (0.18,2.11) |
| | Health insurance card holder | 0.80 (0.28,2.28) | 1.26 (0.61,2.61) | 0.44 (0.15,1.27) | 0.50 (0.15,1.73) | 0.59 (0.17,2.03) | 0.55 (0.23,1.33) | 1.50 (0.65,3.51) | 0.97 (0.28,3.40) |
| | (ref: no health insurance card) | | | | | | | | |
| | reception centre resident | 0.41 (0.12,1.40) | 0.63 (0.28,1.43) | 0.46 (0.15,1.45) | 0.33 (0.10,1.16) | 0.23 (0.08,0.66) ** | 0.52 (0.19,1.47) | 0.77 (0.36,1.67) | 0.35 (0.12,1.01) |
| | (ref: accom. centre resident) | | | | | | | | |
| | urban setting (ref: rural setting) | 1.01 (0.35,2.92) | 1.08 (0.60,1.99) | 0.93 (0.33,2.66) | 1.37 (0.52,3.63) | 1.64 (0.57,4.78) | 0.91 (0.41,2.02) | 1.21 (0.58,2.57) | 1.35 (0.52,3.52) |

FMI = fraction of missing information, SSS = subjective social status, HRQoL = health-related quality of life, PHQ2 = patient health questionnaire 2-item version, GAD2 = generalised anxiety disorder 2-item version

in the choice (OR: 0.34, 95%CI: 0.12,1.02) and communication domains (OR:0.41, 95%CI: 0.17,1.01), although these effects are only borderline significant.

### 3.5. Model fit

The Fraction of Missing Information (FMI) ranged between 25.0% and 46.0% in the fully adjusted, imputed models, indicating the variance attributable to missing data was moderate.

P-values of the F-test ranged from p<0.001 to 0.199 (Table 4), indicating that the variables included in the full models were generally well suited to assessing responsiveness outcomes. Only two models, namely for the confidentiality (p = 0.129) and cleanliness (p = 0.199) domains, did not have sufficient certainty of model fit at the p≤0.05 level.

## 4. Discussion

Overall, over three-quarters of ASR reported "(very) good" responsiveness of the health system in Germany, but rating differs markedly between domains, with choice and timeliness domains receiving particularly poor results. Inequalities in combined responsiveness could be observed in particular in relation to health status factors, with lower responsiveness reported by those with worse mental health, worse HRQoL and a longstanding illness. Being young and having a South Asian nationality emerged as important socio-demographic predictors of low combined responsiveness, while inequalities related to structural factors emerged only in particular responsiveness domains.

Participants from both reception centres and regional accommodation centres reported low scores for the choice and timeliness domains. These, together with the cleanliness domain, represent "client orientation" domains, as opposed to the "respect for persons" domains of respect, communication, autonomy and confidentiality [33]. Features of the health system that lead to delays and a lack of choice in patient care may be expected in the setting of reception centres, where a single walk-in clinic often provides care for many residents [34]. However, these results are unexpected for ASR residing in regional accommodation centres, as they have formal access to regular health service structures. Further research is required to understand why these individuals reports poor choice and timeliness in order to improve the responsiveness of care for this population.

The client-orientation domains were rated comparatively well, indicating the efforts made by frontline staff to continue the same quality of services despite the unusual care setting and the potential challenges in the healthcare encounter. However, communication proved to be a difficult issue, coming in third from last compared to other domains. This may be expected given the pervasive issue of language barriers in the provision of health services for ASR [34]. However, it is unclear whether this rating relates to the lack of adequate interpreting services or the communication abilities of the physicians.

Our study found inequalities in responsiveness related to health status factors. These may result in horizontal inequities in health system access if low responsiveness impedes further engagement with the healthcare system: "Responsiveness that is systematically worse for certain social groups with the same or greater needs than other social groups could lead to inequities in access" (9). Thus, individuals with pre-existing conditions and illnesses, who may be most in need of responsive health services, are currently being sold short. This finding is particularly striking given the common assumption that responsiveness ratings improve with continued engagement and interaction with the health system [33]. Further research is required to understand the particular reasons for worse responsiveness rating for individuals in poor health and the development of system-based interventions to improve quality of care for this patient group.

Furthermore, lower responsiveness can also be observed for younger individuals and those with a South Asian nationality. This may indicate discrimination against these population groups within the health system. However, the observed differences may also be grounded in different expectations of what the health system should be able to deliver along responsiveness domains. Responsiveness has been recognized as a health system outcome which is strongly influenced by differential rating according to underlying health system expectations [19].

Thus, further studies should investigate whether the observed inequalities remain stable once differential expectations have been adjusted for, for example through the use of vignettes.

The factors found to be relevant in predicting combined "(very) good" responsiveness were not entirely consistent across different domains. This highlights the added value of creating a combined responsiveness score. While in some cases (e.g. age and high GAD2 score), relationships with responsiveness could be observed in several domains, in other cases (e.g. South Asian nationality and HRQoL), important effects may not have been picked up by analysing only the individual domains. Consistently low scores in each domain, although not necessarily relevant by themselves, resulted in strong predictions in the combined model. However, there is no commonly accepted and validated method of combining results across responsiveness domains, and the method used for this analysis should be assessed for reliability and validity in further research.

While there is no agreed cut-off for the level of responsiveness health systems should achieve, comparison with a recent population-based survey on responsiveness among German patients provides a good reference for putting these figures in context [35]. German patients report responsiveness ratings between 54.2%-96.2% across domains, with higher responsiveness ratings than our study in every domain. This comparison is suggestive of inequalities in responsiveness between ASR and the resident German population, and should be further explored in future studies.

A key feature of this study is that it is the first to apply the WHO Responsiveness survey in a diverse ASR population [36]. The comprehensiveness of the WHO Responsiveness instrument makes it a good tool to survey patient-rated quality of care and identify areas for improvement. The analysis of individual domains revealed complex patterns, reflecting both the heterogeneity of the ASR population and the complexity of the responsiveness concept. This study cannot comprehensively assess all dimensions encompassed in the responsiveness concept, but gives an overview of potential issues and topics for further inquiry. Further research into specific domains, including the use of qualitative methods, should be conducted to understand reasons behind responsiveness answers and differences in particular population groups. For example, recently published qualitative research into the "patient journey" of ASR following arrival in Germany shows that positive responsiveness ratings may be attributed to a high level of generalised trust in the healthcare profession [37].

This study benefits from a population-based, random sampling approach, a personal data collection approach [15, 18] and rigorous weighting and imputation prior to analysis. A limitation of this study was a result of our data collection approach: in order to keep the questionnaire short only one set of responsiveness questions was included for all healthcare sectors. This makes it impossible to provide responsiveness results for ambulatory and in-patient care separately, as was done by the WHO. Future studies should consider asking for the sector which was being rated if the inclusion of two separate questionnaires proves difficult.

A further limitation is the potential non-response bias introduced by both the linguistic diversity and the literacy of the sample population. Non-response may also be an issue for the analysis of responsiveness itself, as individuals which did not visit a healthcare provider in the last 12 months are excluded from the measurement by design. As has been noted previously, this may lead to an upward bias in results if individuals who do not come into contact with the health system due to low responsiveness of care are not captured [9].

Finally, due to the different requirements of the imputation for the weighting and the analysis steps, two imputations were performed. This may mean that weights were calculated on the basis of slightly different data than was used for the final analysis. However, given the relatively small effects of the weighting approach we do not expect that this substantially affected the results of the final models.

## 5. Conclusion

This analysis found important differences in responsiveness related to health, socio-demographic and structural factors both in combined responsiveness and in individual domains among a diverse population of ASR. Inequalities related to health status factors are particularly concerning given the potential implications for equity in access to health care. The WHO responsiveness survey proved to be a useful concept to quantify quality of care among ASR and map existing inequalities from the patient perspective. This study therefore makes a novel contribution to the current literature on Universal Health Coverage for ASR by introducing aspects of the quality of care to existing analyses of eligibility and accessibility. Further research is needed to understand the relationships between responsiveness and the socio-demographic, health and structural factors explored in this analysis in more detail and to derive relevant system-level interventions to improve quality of care.

## Supporting information

**S1 Fig. STROBE chart representing sampling, data collection analysis process and number of individuals in reception centres and accommodation centres at each stage.**
(DOCX)

**S1 Table. Categorisation of educational score from survey items relating to school and continuing education.**
(DOCX)

**S2 Table. Outcomes of logistic models of combined responsiveness with health factors and apriori confounders.**
(DOCX)

**S3 Table. Outcomes of logistic models of combined responsiveness with socio-demographic factors and apriori confounders.**
(DOCX)

**S4 Table. Outcomes of logistic models of combined responsiveness with structural factors and apriori confounders.**
(DOCX)

**S1 Text. Additional details of the methodological approach.**
(DOCX)

## Author Contributions

**Conceptualization:** Louise Biddle.

**Data curation:** Louise Biddle.

**Formal analysis:** Louise Biddle, Judith Wenner, Kayvan Bozorgmehr.

**Funding acquisition:** Kayvan Bozorgmehr.

**Methodology:** Louise Biddle, Judith Wenner, Kayvan Bozorgmehr.

**Project administration:** Louise Biddle.

**Supervision:** Kayvan Bozorgmehr.

**Visualization:** Louise Biddle.

**Writing – original draft:** Louise Biddle.

**Writing – review & editing:** Judith Wenner, Kayvan Bozorgmehr.

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
