## [Decision Letter · Decision Letter 0]

18 Jul 2022

PGPH-D-22-00974

Inequalities in health system responsiveness among asylum seekers and refugees: A population-based, cross-sectional study in Germany

Dear Dr. Biddle,

Thank you for submitting your manuscript to PLOS Global Public Health. After careful consideration, we feel that it has merit but does not fully meet PLOS Global Public Health’s publication criteria as it currently stands. Therefore, we invite you to submit a revised version of the manuscript that addresses the points raised during the review process.Please submit your revised manuscript by Aug 17 2022 11:59PM. If you will need more time than this to complete your revisions, please reply to this message or contact the journal office at globalpubhealth@plos.org. Please include the following items when submitting your revised manuscript:

We look forward to receiving your revised manuscript.

Kind regards,

Ejemai Eboreime, MD, MSc, PhD

Academic Editor

Journal Requirements:

Additional Editor Comments (if provided):

Reviewers' comments:

Reviewer's Responses to Questions

**Comments to the Author**

1. Does this manuscript meet PLOS Global Public Health’s publication criteria? Is the manuscript technically sound, and do the data support the conclusions? The manuscript must describe methodologically and ethically rigorous research with conclusions that are appropriately drawn based on the data presented.

Reviewer #1: Yes

Reviewer #2: Yes

Reviewer #3: Partly

2. Has the statistical analysis been performed appropriately and rigorously?

Reviewer #1: I don't know

Reviewer #2: Yes

Reviewer #3: Yes

3. Have the authors made all data underlying the findings in their manuscript fully available (please refer to the Data Availability Statement at the start of the manuscript PDF file)?

Reviewer #1: Yes

Reviewer #2: Yes

Reviewer #3: Yes

4. Is the manuscript presented in an intelligible fashion and written in standard English?

Reviewer #1: Yes

Reviewer #2: Yes

Reviewer #3: Yes

5. Review Comments to the Author

Reviewer #1: Important topic

Need to clarify 15 vs 18mths for entitlements to care as the text says both.

Good description of limitations and recognition the results may not reflect a deeper problem with access - I would add this to the abstract

I am not able to comment on the data analysis for this paper as I am not sufficiently trained in statistics. It appears ok to me but I am hesitant to comment.

Reviewer #2: This is an importand and well written article withe sound methodology. My comments are.

1. The discusion is well written but could be shortened

2. The limitations of survey data in terms of its ability to reflect facts could be pointed out. This could be tied to those with long term chronic illness and also anxiety feeling they are not recieving responsive care.

Reviewer #3: Reviewer’s Comments

Summary

Thank you for the opportunity I have to provide a review for this important article on inequalities in health system responsiveness among ASR in Germany. This article funded by BMBF focuses on assessing the quality of care among ASRs using the concept of responsiveness. The authors’ results and conclusions are quite informative and good for decision making by the ministry.

Cover page

•The title clearly depicts the research that was carried out with focus on the outcome.

•Each author’s contribution was clearly stated, and corresponding author’s contact is clear.

Abstract

•The objective was to search if there were inequalities in health care among the population of studied and the summary on inequality findings was clearly written in the conclusion.

•It is an interesting read for those who want to learn more about responsiveness in health systems.

Introduction

•This aspect was well written and introduced the readers to the need to want to find out what the authors found out.

•However, the summary from reference 4 does not fully itemize the 5 goals of the Lancet commission, whose report was in the article. The authors can consider reviewing the sentence in relations to the article they quoted.

•Readers were effectively introduced to the concept of responsiveness- this is very good

•The objectives are well stated at the end of the introduction

•Although I am of the opinion that the authors could have moved the last sentence in introduction to conclusion.

Methods and Materials

•I would advise the authors to provide the setting of the study to the readers of the journal. Not everyone knows the Germany’s third largest federal state, its population and the population of the ASRs.

•Sample size calculation was not stated, only data collection method was mentioned (“door to door approach”). Another researcher might not be able to repeat this method because of lack of guidance on how sampling size was determined.

•The abstract stated that simple random sampling was done, but there is NO section on sampling technique in the methods and material section.

•Data management was thorough and quite informative.

Results

•Response rate was low, and it is good that the authors stated it. However, it has been compensated for in the statistical analysis.

•The results of the 2 objectives of the study were clearly stated.

Discussion

•The discussion is well written and provides the necessary reasoning behind the results of the study

oEspecially when the researchers found a negative results in their study, they were able to provide probable reason for it

Conclusion

•Provides answers to the two aims of the study

•The recommendations made by the authors is appropriate for the study

References

•The references are complete and well cited

6. PLOS authors have the option to publish the peer review history of their article (what does this mean?). If published, this will include your full peer review and any attached files.

**Do you want your identity to be public for this peer review?** For information about this choice, including consent withdrawal, please see our Privacy Policy.

Reviewer #1: No

Reviewer #2: No

Reviewer #3: **Yes: **Dr Adebayo Peter Adewuyi

---

## [Decision Letter · Decision Letter 1]

1 Sep 2022

Inequalities in health system responsiveness among asylum seekers and refugees: A population-based, cross-sectional study in Germany

PGPH-D-22-00974R1

Dear Mrs. Biddle,

We are pleased to inform you that your manuscript 'Inequalities in health system responsiveness among asylum seekers and refugees: A population-based, cross-sectional study in Germany' has been provisionally accepted for publication in PLOS Global Public Health.

Best regards,

Ejemai Eboreime, MD, MSc, PhD

Academic Editor

Reviewer Comments (if any, and for reference):

Reviewer's Responses to Questions

**Comments to the Author**

1. If the authors have adequately addressed your comments raised in a previous round of review and you feel that this manuscript is now acceptable for publication, you may indicate that here to bypass the “Comments to the Author” section, enter your conflict of interest statement in the “Confidential to Editor” section, and submit your "Accept" recommendation.

Reviewer #2: All comments have been addressed

Reviewer #3: All comments have been addressed

2. Does this manuscript meet PLOS Global Public Health’s publication criteria? Is the manuscript technically sound, and do the data support the conclusions? The manuscript must describe methodologically and ethically rigorous research with conclusions that are appropriately drawn based on the data presented.

Reviewer #2: Yes

Reviewer #3: Yes

3. Has the statistical analysis been performed appropriately and rigorously?

Reviewer #2: Yes

Reviewer #3: Yes

4. Have the authors made all data underlying the findings in their manuscript fully available (please refer to the Data Availability Statement at the start of the manuscript PDF file)?

Reviewer #2: Yes

Reviewer #3: Yes

5. Is the manuscript presented in an intelligible fashion and written in standard English?

Reviewer #2: Yes

Reviewer #3: Yes

6. Review Comments to the Author

Reviewer #2: The article is well written, the analysis is rigourous, and the information is helpful.

Reviewer #3: ALL comments have been addressed. I want to thank the authors for deciding to address the comments because it will give the readers an opportunity to fully understand the work done in this manuscript. Well done

7. PLOS authors have the option to publish the peer review history of their article (what does this mean?). If published, this will include your full peer review and any attached files.

**Do you want your identity to be public for this peer review?** For information about this choice, including consent withdrawal, please see our Privacy Policy.

Reviewer #2: No

Reviewer #3: **Yes: **ADEBAYO PETER ADEWUYI
